# Implementation and Early Outcomes of an Antimicrobial Stewardship Program in South Korea

**DOI:** 10.3390/antibiotics14080834

**Published:** 2025-08-17

**Authors:** Kyeong Min Jo, Tae-Hoon No

**Affiliations:** Department of Infectious Diseases, Inje University Haeundae Paik Hospital, Busan 48108, Republic of Korea; tae-hoon.no@paik.ac.kr

**Keywords:** antimicrobial stewardship program, infection, antibiotics

## Abstract

**Background**: Antimicrobial stewardship programs (ASPs) are essential for promoting the rational use of antibiotics and combating resistance. In South Korea, implementation has recently accelerated, but real-world data on short-term program performance remain limited. This study evaluated the early outcomes of a newly launched ASP at a tertiary hospital. **Methods**: This retrospective, single-center study analyzed ASP activity from January to April 2025. Interventions included prospective audit and feedback for restricted antibiotics and recommendations for prolonged antibiotic prescriptions (≥14 days). The primary outcome was the monthly rejection rate of restricted antibiotics. Secondary outcomes included days of therapy (DOT) for restricted antibiotics and the acceptance rate of interventions for prolonged prescriptions. **Results**: The monthly rejection rate of restricted antibiotics remained stable between 3.65% and 4.68%. Although the DOT values did not show statistical significance, they demonstrated a moderate inverse correlation with the rejection rate (Pearson’s r = –0.868, *p* = 0.132). Among 826 prolonged prescriptions, 513 (62.1%) received ASP intervention. Acceptance of recommendations increased over time, from 67.0% in January to 82.5% in April. Interventions were primarily based on insufficient evidence of infection or inappropriate antibiotic selection. **Conclusions**: The newly implemented ASP demonstrated feasibility and early impact in improving antibiotic oversight. Despite the short observation period, the findings suggest a positive influence on prescribing practices. Longer-term studies are needed to evaluate sustained clinical outcomes and broader applicability in diverse healthcare settings.

## 1. Introduction

Antibiotic resistance (ABR) is a natural process by which microorganisms develop resistance to antibiotics, increasingly driven by their overuse in healthcare, agriculture, and the environment [1,2,3,4,5]. To combat widespread antibiotic misuse in both community and hospital settings, antimicrobial stewardship programs (ASPs) have been encouraged, particularly within healthcare facilities [6,7,8]. As the term “antimicrobial stewardship” has been widely used without a universally agreed definition, its interpretation has varied across settings, shaped by local practices and perspectives. This diversity continues to influence how the concept is discussed and understood [9]. ASPs aim to optimize antimicrobial prescribing practices through various interventions, including formulary restrictions, prospective audit and feedback, and prescriber education. Such programs may improve clinical outcomes, reduce antibiotic consumption, and lower resistance rates [10,11,12,13,14,15]. The core strategies of ASPs include formulary restriction, pre-authorization of high-risk antibiotics, prospective audit with feedback, nurse education and prescriber education [16,17,18,19,20].

In South Korea, the implementation of ASPs has gained increasing attention in recent years, driven in part by national-level initiatives and funding support from the Korea Disease Control and Prevention Agency (KDCA) [21]. However, real-world data evaluating the clinical impact of ASPs in Korean hospitals remain limited, especially with respect to short-term program performance and physician adherence.

At our institution in the Republic of Korea, a formal ASP was launched in November 2024 to improve antibiotic oversight, with a focus on restricted antimicrobial agents. The program involved daily reviews of restricted antibiotic prescriptions following multidisciplinary collaboration among infectious disease specialists, pharmacists, and skilled nurses. This study aimed to assess the early outcomes of the newly implemented ASP by analyzing monthly changes in restricted antibiotic rejection rates, days of therapy (DOT), and intervention acceptance for prolonged antibiotic use over a 4-month period.

## 2. Results

During the 4-month evaluation period from January to April 2025, the total number of restricted antibiotic prescriptions ranged from 449 to 640 per month. Approval and rejection patterns of each restricted antibiotic are summarized in Appendix A. A most frequently used restricted antibiotics list is summarized in Appendix A and Appendix A. The monthly rejection rate ranged from 3.65% to 4.68%, showing moderate fluctuations without a clear directional pattern (Figure 1).

We analyzed monthly DOT for all restricted antibiotics to assess the impact of ASP restrictions on actual antimicrobial use. The total DOT for the restricted agents decreased from 125.1 in January to 97.0 in February and then increased slightly to 115.9 in March and 108.7 in April, showing fluctuating values over the observation period (Figure 2).

From January to April 2025, 826 patients received prolonged antibiotic therapy (≥14 days). Among them, 513 (62.1%) required intervention by an ASP team. Interventions were recommended primarily when prolonged antibiotic use lacked clear justification, such as the absence of clinical or microbiological evidence of ongoing infection or inappropriate antibiotic selection not aligned with identified pathogens.

The number of cases requiring intervention increased from 86 in January to 177 in April. Correspondingly, the acceptance rate of ASP recommendations improved. The acceptance rates in January, February, March, and April were 67.0% (58 accepted out of 86 interventions), 65.5%, 72.5%, and 82.5%, respectively (Figure 3). Monthly DOT by restricted antibiotics are summarized in Appendix A.

## 3. Discussion

This study evaluated the short-term outcomes of a newly implemented antimicrobial stewardship program (ASP) at a tertiary hospital in South Korea. The four-month observation period (January to April 2025) reflects the early implementation phase of the ASP pilot project at this hospital. This initial timeframe was selected to assess the feasibility and preliminary outcomes of ASP interventions in real-world clinical practice. Over the first four months of implementation, we observed consistent control of restricted antibiotic prescriptions, with a monthly rejection rate maintained between 3.65% and 4.68%. Although the total days of therapy (DOT) for restricted antibiotics did not decrease significantly, the observed pattern may indicate a potential reduction in broad-spectrum antibiotic use due to the stewardship intervention.

Importantly, the ASP team also targeted prolonged antibiotic prescriptions (≥14 days), a frequently overlooked area in routine stewardship practice. Among 826 total prolonged prescriptions, 513 cases (62.1%) were intervened upon by the ASP team, with a notable increase in intervention acceptance rates from 55% in January to 72% in April. These interventions were primarily based on the absence of clear evidence of ongoing infection or the inappropriate selection of agents relative to culture results. The increase in acceptance may reflect growing familiarity and trust in the stewardship process among clinicians.

These findings highlight the feasibility and early effectiveness of ASPs even during the initial phases of implementation, particularly in institutions with limited prior experience. The relatively stable rejection rate of restricted antibiotics suggests that ASP-led gatekeeping can be performed consistently when supported by a multidisciplinary team and electronic prescribing systems. Furthermore, the acceptance of stewardship recommendations regarding prolonged antibiotic use emphasizes the importance of clinically grounded criteria and ongoing communication between stewardship teams and prescribing physicians.

Our findings align with previous studies that demonstrated early ASP benefits in reducing unnecessary antimicrobial use and promoting rational prescribing behavior [10,11,12,22,23]. However, unlike large multi-center studies, our analysis provides practical insights into short-term ASP deployment under real-world conditions in a single-center context.

There are several limitations to this study. The observation period was limited to four months, which may not fully capture seasonal variations or long-term program sustainability. However, even within this relatively short period, we observed early signs of adoption and acceptance of ASP interventions. These initial findings highlight the feasibility of implementing stewardship activities in routine clinical workflows and warrant extended follow-up to assess long-term outcomes.

We did not assess patient-level clinical outcomes, such as infection recurrence, adverse events, or mortality. The current ASP team at our institution remains in its early stages, consisting of only two infectious disease physicians. Due to limited human resources, it was not feasible to perform a comprehensive, patient-level review of all antibiotic prescriptions to assess indications or infection status in detail. To address this limitation, the ASP team recommended formal infectious diseases consultations for cases where the appropriateness of prolonged antibiotic use was uncertain. As the hospital’s ASP program expands and additional personnel—such as infection control nurses or pharmacists—are incorporated, more granular assessments of infection-related indications and targeted interventions will become feasible in future studies.

The study was conducted at a single institution, which may limit the generalizability of the findings to other settings with different resources or antimicrobial stewardship infrastructures. However, as a real-world implementation of an ASP in a typical secondary care hospital, this study reflects practical challenges and achievable impacts in routine clinical practice. The framework and outcomes presented here may serve as a foundation for future multi-center research aimed at confirming broader applicability.

While we categorized and monitored restricted antibiotics as part of our institutional ASP policy, we did not specifically align them with the WHO AWaRe (Access, Watch, Reserve) classification in the current analysis. However, the majority of the agents requiring approval in our institution, such as meropenem, vancomycin, colistin, and linezolid, fall under the ‘Watch’ or ‘Reserve’ categories, indicating high potential for resistance and strong need for stewardship oversight. Future studies may benefit from explicitly incorporating AWaRe-based categorization to better align local stewardship interventions with global antibiotic policy frameworks.

Although the dataset included information on individual restricted antibiotics, we did not perform a component-specific analysis due to the limited scope of the study. This remains an area for future investigation.

Despite these limitations, this study offers meaningful evidence supporting the early impact and acceptability of ASP interventions targeting both restricted and prolonged antibiotic use. Continued monitoring, expanded outcome measures, and multi-center collaboration are needed to further validate the long-term clinical and epidemiological benefits of ASPs in similar healthcare environments.

In Korea, securing ID-trained personnel for ASP remains challenging due to low job satisfaction and high turnover, prompting government-led initiatives and the development of structured education programs to enhance stewardship capacity among non-ID clinicians [24,25,26,27,28,29,30,31]. Therefore, this study holds particular significance by presenting real-world data on ASP implementation in Korea, while also shedding light on the structural and operational challenges faced by under-resourced ASP teams. Such findings may inform future policy adjustments and workforce development strategies.

## 4. Material and Methods

### 4.1. Study Design and Setting

This single-center, retrospective, cross-sectional study was conducted using data collected between January 2025 and April 2025 at a tertiary care hospital in the Republic of Korea. The ASP was implemented in collaboration with a multidisciplinary team comprising infectious disease specialists, clinical pharmacists, and microbiology laboratory personnel. Core strategies included prospective audit and feedback, pre-authorization of restricted antimicrobials, and communication with prescribers.

### 4.2. ASP Intervention

The ASP interventions targeted all prescriptions of restricted antibiotics across all departments within the hospital, including internal medicine, general surgery, pediatrics, obstetrics and gynecology, orthopedics, neurosurgery, urology, and others. Restricted antibiotics were defined by the hospital’s formulary committee based on their broad-spectrum activity, high cost, or potential for resistance. These included glycopeptides (e.g., vancomycin), oxazolidinones (e.g., linezolid), carbapenems (e.g., meropenem and imipenem), polymyxins (e.g., colistin), lipopeptides (e.g., daptomycin), and triazole antifungals (e.g., voriconazole and isavuconazole). The list of restricted antibiotics subject to ASP intervention, along with their approval criteria, is summarized in Appendix A.

ASP interventions were primarily conducted via the hospital’s electronic medical record (EMR) system using an internal messaging function. When a restricted antibiotic was prescribed, an automatic consultation request was triggered and sent to the ASP team. Upon review, the ASP team’s decision—either approval or a recommendation for adjustment—was communicated directly to the prescribing physician through the EMR. If the recommendation was not accepted, follow-up messages were sent, and direct communication between the infectious disease specialist and attending physician was initiated in some cases. The acceptance rate was calculated as the percentage of restricted antibiotic prescriptions that were approved by the ASP team among total prescription requests. The rejection rate referred to the percentage of requests that were denied. Days of therapy (DOT) was defined as the number of days a patient received a specific antibiotic, with one DOT counted per agent per day, regardless of dose or frequency.

### 4.3. Study Outcomes

The primary outcome was the monthly rejection rate of the restricted antibiotic prescriptions, calculated as the proportion of restricted antibiotic prescriptions not approved by the ASP team. This metric served as a proxy indicator for appropriateness of prescribing and effectiveness of ASP. Secondary outcomes reflected the qualitative implementation of ASP. Specifically, we assessed the number of ASP interventions targeting prolonged antibiotic use, defined as the administration of the same antibiotic agent for ≥14 days. Approval and rejection patterns of antibiotic agents were analyzed to evaluate the intensity of stewardship oversight for each restricted drug. Department-level acceptance rates were reviewed to assess variability in compliance. Overall antibiotic use was summarized using days of therapy (DOT) to provide context for the overall pattern of prescriptions. The acceptance rate (%) was defined as the proportion of ASP interventions that were accepted by the prescribing physician, calculated by dividing the number of accepted interventions by the total number of ASP recommendations per month.

### 4.4. Statistical Analysis

All outcome data were retrospectively extracted from the hospital’s electronic medical records and the ASP intervention logs. Monthly data were analyzed using descriptive statistics, chi-square tests, and linear regression analyses, as appropriate. Statistical analyses were conducted using IBM SPSS Statistics version 26.0 (IBM Corp., Armonk, NY, USA).

### 4.5. Ethical Concerns

This study was approved by the Institutional Review Board (IRB) of Inje University Haeundae Paik Hospital (IRB No. 2025-05-020).

## 5. Conclusions

This single-center study demonstrates the feasibility and early impact of an Antimicrobial Stewardship Program (ASP) implemented at a tertiary hospital in South Korea. The program maintained stable rejection rates for restricted antibiotics and showed increasing acceptance of interventions for prolonged antibiotic prescriptions. Over a four-month period, we observed a modest but consistent decrease in DOT for restricted antibiotics, suggesting a positive impact of stewardship interventions. Although the changes in antibiotic use did not reach statistical significance, they may reflect potential clinical relevance; however, the limited sample size and study duration restrict definitive conclusions. A longer-term follow-up study is warranted to more accurately evaluate the true impact of ASP interventions. These findings suggest that structured ASP efforts can promote more appropriate antibiotic use, even in the early phases of implementation. Future studies are needed to assess the long-term sustainability and broader clinical benefits of ASPs across diverse healthcare settings.

## Figures and Tables

**Figure 1 antibiotics-14-00834-f001:**
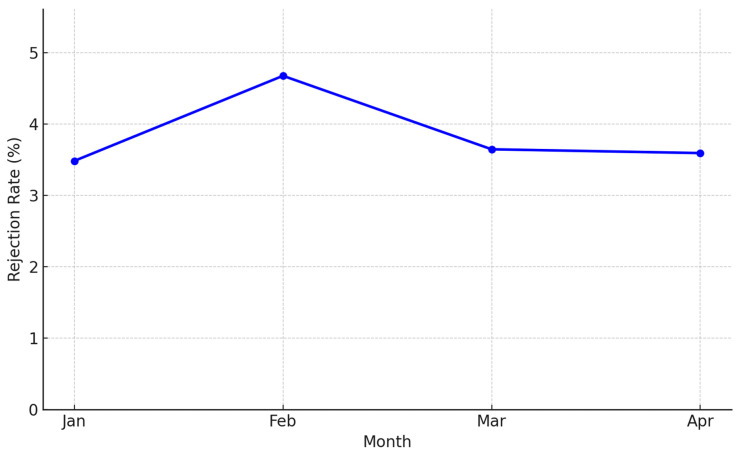
Monthly rejection rate of restricted antibiotics.

**Figure 2 antibiotics-14-00834-f002:**
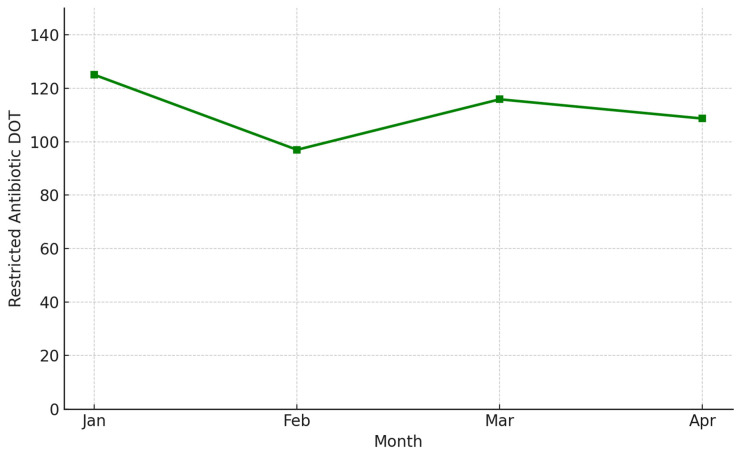
Monthly Days of Therapy (DOT) of restricted antibiotics.

**Figure 3 antibiotics-14-00834-f003:**
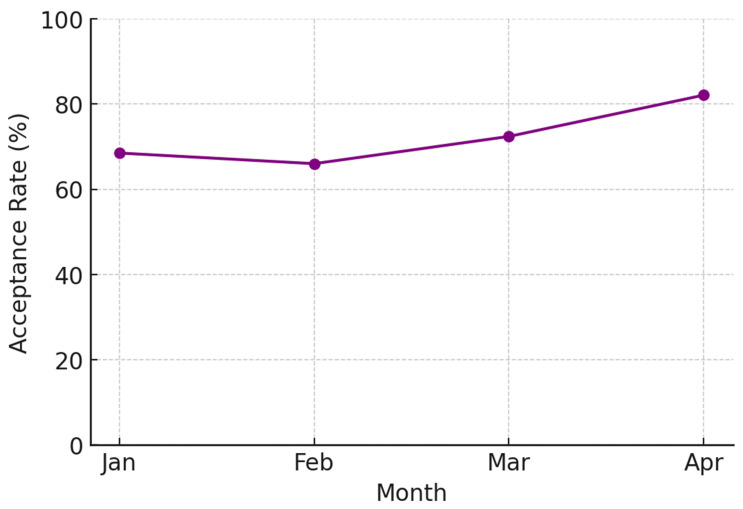
Monthly acceptance rate of interventions for prolonged antibiotics use.

## Data Availability

The original contributions presented in this study are included in the article/Appendix A. Further inquiries can be directed to the corresponding author(s).

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
