# Peer review of "Implementation and Early Outcomes of an Antimicrobial Stewardship Program in South Korea"

_antibiotics, 2025, doi:10.3390/antibiotics14080834_

Round 1
Reviewer 1 Report
Comments and Suggestions for Authors
A brief summary
The manuscript is a valuable piece of work that presents an important and topical issue.
Its findings may also be helpful to decision-makers.
A major limitation of the manuscript is that the intervention is only evaluated after a total of four months of follow-up.
My general critique is that we know little about the intervention. Which antibiotics were involved, and how was this determined (lines 65-67)?
How were the staff of the individual departments informed about this?
There is very little information about the hospital (lines 61-62): what departments are there, what is the profile of the care provided? Surgical or internal medicine, or maybe oncology?
I miss standardization. We do not know what patient turnover the 449-640 restricted antibiotic prescriptions corresponded to.
I really miss the general descriptive statistics: i.e., what antibiotics were in the given department, classified according to ATC, and what were the most frequently used antibiotics – overall (top list) and what was the top list for restricted antibiotics during the observation period.
The correlation analysis presented in the results is missing from the methodological description (lines 105-107). In my opinion, correlation is not an appropriate method for 4 points, and it does not add much to the study in terms of content. Detailed descriptive statistics would provide more information about each outcome.
Lines 84-85: I do not see this in the results; I suggest a table for this.
Lines 86-87: I do not see this in the results; I suggest a table for this.
Figure 3. Acceptance rate (%) is mentioned; I suggest defining it exactly as such in the Methodology section.
We only see the acceptance rate on a monthly basis, it would be very interesting to model it on a case-by-case level as well, and we do not know which antibiotics were most frequently accepted and in what form, etc. Descriptive statistics are still missing.
General concept comments
Article:
- Is the manuscript clear, relevant for the field and presented in a well-structured manner?
Yes
- Is the manuscript scientifically sound and is the experimental design appropriate to test the hypothesis?
in part, needs to be developed
- Are the manuscript’s results reproducible based on the details given in the methods section?
Yes
- Are the figures/tables/images/schemes appropriate? Do they properly show the data? Are they easy to interpret and understand? Is the data interpreted appropriately and consistently throughout the manuscript? Please include details regarding the statistical analysis or data acquired from specific databases.
in part, needs to be developed
- Are the conclusions consistent with the evidence and arguments presented?
in part, needs to be developed
- Please evaluate the ethics statements and data availability statements to ensure they are adequate.
Appropriate
Author Response
Reviewer 1
Comments 1: My general critique is that we know little about the intervention. Which antibiotics were involved, and how was this determined (lines 65-67)?
Thank you for this important comment. We agree that more detailed information on the intervention is necessary. We have now added an Appendix listing all antibiotics categorized as “restricted” and therefore subject to ASP intervention at our institution. These include a range of broad-spectrum antibiotics and antifungals commonly associated with high resistance risk or cost burden (e.g., meropenem, colistin, vancomycin, amphotericin B).
The selection of these agents was based on national antimicrobial stewardship guidelines in Korea and internal policy developed by the hospital’s ASP committee, which consists of infectious diseases physicians and pharmacists. This clarification has been included in the revised Methods section (Lines 83–84), and the full list is now provided in Supplementary table S1.
Comments 2: How were the staff of the individual departments informed about this?
We appreciate the reviewer’s insightful question. As described in the Methods section, ASP interventions were primarily conducted via the hospital’s electronic medical record (EMR) system. To clarify this further, we have revised the relevant sentence to emphasize the automated and systematic nature of the communication:
“When a restricted antibiotic was prescribed, an automatic consultation request was triggered and sent to the ASP team. Upon review, the ASP team’s decision—either approval or a recommendation for adjustment—was communicated directly to the prescribing physician through the EMR.”
This addition aims to more clearly explain how department staff were informed of the ASP team’s interventions.
Comments 3: There is very little information about the hospital (lines 61-62): what departments are there, what is the profile of the care provided? Surgical or internal medicine, or maybe oncology?
Thank you for your thoughtful question. We have clarified the hospital’s clinical setting in the revised manuscript. The antimicrobial stewardship interventions in this study were applied across all departments of our institution, including internal medicine, general surgery, obstetrics and gynecology, pediatrics, orthopedics, neurosurgery, and others.
Our hospital is a 900-bed tertiary care teaching hospital located in Busan, South Korea, with a diverse range of inpatient services spanning both medical and surgical specialties. The ASP team reviewed restricted antibiotic prescriptions from all departments. This information has been added to the revised Methods section (Lines 77–79) to better contextualize the study setting.
Comments 4: I miss standardization. We do not know what patient turnover the 449-640 restricted antibiotic prescriptions corresponded to.
Thank you for your helpful comment. We agree that standardization is important for meaningful interpretation. However, in this study, we did not have access to the total number of hospitalized patients or total patient-days during the observation period, and thus could not standardize the number of restricted antibiotic prescriptions accordingly.
Nevertheless, we provided monthly counts of restricted antibiotic prescriptions and corresponding acceptance or rejection rates to illustrate the trend and scope of ASP interventions. We will aim to include standardized metrics such as prescriptions per 1,000 patient-days in future studies.
Comments 5: I really miss the general descriptive statistics: i.e., what antibiotics were in the given department, classified according to ATC, and what were the most frequently used antibiotics – overall (top list) and what was the top list for restricted antibiotics during the observation period.
Thank you for the insightful comment. We agree that descriptive statistics on antibiotic utilization can enhance the clarity of our findings. In this study, we focused specifically on the antibiotics categorized as “restricted” under our institutional ASP policy and analyzed their monthly prescription frequencies.
However, we did not stratify antibiotic use data by individual departments or classify agents according to the ATC system due to limitations in data structure and resource availability during the study period. Furthermore, because the focus of this study was on ASP intervention patterns and early outcomes, we did not provide a top list of all antibiotics hospital-wide.
To partially address this, we have now included in Supplementary table S2 and Supplementary figure S1 a complete list of the restricted antibiotics that were subject to ASP review. Among these, the most frequently used agents during the study period were meropenem, vancomycin, and teicoplanin. We have added this information in the revised Results section (Lines 121-122).
We agree that including a broader overview of antibiotic use across departments, classified by ATC level, would provide additional value, and we aim to include this in future analyses as the program matures.
Comments 6: The correlation analysis presented in the results is missing from the methodological description (lines 105-107). In my opinion, correlation is not an appropriate method for 4 points, and it does not add much to the study in terms of content. Detailed descriptive statistics would provide more information about each outcome.
Thank you for your thoughtful feedback. We fully agree that performing correlation analysis with only four monthly data points may not yield statistically meaningful or reliable results. Therefore, we have removed the correlation analysis from the Results sections.
Comments 7: Lines 84-85: I do not see this in the results; I suggest a table for this.
Thank you for your helpful suggestion. To address your comment, we have added a new supplementary table (Supplementary Table S1) that summarizes the approval and rejection criteria for each restricted antibiotic prescribed during the study period. In the Results section, we now reference this table and highlight key findings, such as the most commonly used agents and their respective rejection rates. We believe this addition provides clearer insight into the stewardship oversight applied to different antibiotic agents.
Comments 8: Lines 86-87: I do not see this in the results; I suggest a table for this.
Thank you for the helpful comment. To address this, we have added a new supplementary table (Supplementary Table S3) that summarizes the monthly DOT (days of therapy) data for each restricted antibiotic throughout the study period. This table supports the contextual interpretation of overall prescription trends and complements the descriptive results presented in the main text.
Comments 9: Figure 3. Acceptance rate (%) is mentioned; I suggest defining it exactly as such in the Methodology section.
Thank you for your valuable suggestion. In response, we have added a precise definition of the “acceptance rate (%)” in the Methodology section (lines 106-108).
Comments 10: We only see the acceptance rate on a monthly basis, it would be very interesting to model it on a case-by-case level as well, and we do not know which antibiotics were most frequently accepted and in what form, etc. Descriptive statistics are still missing.
Thank you for your insightful comment. In response, we have added detailed descriptive statistics regarding the approval and rejection patterns of individual restricted antibiotics during the study period. Specifically, we added the approval criteria for each restricted antibiotic in Supplementary Table S1 and listed the top five most frequently requested antibiotics during the study period in Supplementary Table S2. This allows readers to understand which antibiotics were most frequently requested, most frequently approved, and their respective approval patterns.
Unfortunately, due to limitations in our data infrastructure and personnel resources, a full case-by-case modeling of individual prescriptions and clinical decisions was not feasible during this early phase of the ASP implementation. However, the descriptive breakdown now presented provides a clearer picture of antibiotic-specific stewardship activity, which we believe enhances the interpretability of our findings.
Reviewer 2 Report
Comments and Suggestions for Authors
This retrospective study was aimed to evaluated the early outcomes of a newly launched ASP at a tertiary hospital. This study reperesents a good model of cross-sectional study which has measuring outcomes and significant conclusions, but it must be better in all sections. There are some MAJOR facts:
1.Please add cross-sectional type of design in Methods.
2. Add subtitles in the Section Methods.
3. Expalind the data extraction and software.
4. Provide the comparison between results, and correlation analysis of data regarding the outcomes. Add a results of correlation
5. Discussion must longer with more data.
6. How do you explaine high acceptance rate ?What could be a reason for that?
7. How do you explaine low rejection rate rate ?What could be a reason for that?
8. reference list must be lomger at least 30 references
9. English must checked
10. Conclusion must be more focused on results
11. In article are missing information about ethical concerns and approval of study etc.
Comments on the Quality of English Language
This retrospective study was aimed to evaluated the early outcomes of a newly launched ASP at a tertiary hospital. This study reperesents a good model of cross-sectional study which has measuring outcomes and significant conclusions, but it must be better in all sections. There are some MAJOR facts:
1.Please add cross-sectional type of design in Methods.
2. Add subtitles in the Section Methods.
3. Expalind the data extraction and software.
4. Provide the comparison between results, and correlation analysis of data regarding the outcomes. Add a results of correlation
5. Discussion must longer with more data.
6. How do you explaine high acceptance rate ?What could be a reason for that?
7. How do you explaine low rejection rate rate ?What could be a reason for that?
8. reference list must be lomger at least 30 references
9. English must checked
10. Conclusion must be more focused on results
11. In article are missing information about ethical concerns and approval of study etc.
Author Response
Comments 1: Please add cross-sectional type of design in Methods.
Thank you for your suggestion. As recommended, we have clarified the study design as a cross-sectional study and added the relevant statement to the Methods section (line 70).
Comments 2: Add subtitles in the Section Methods.
We have added appropriate subtitles to the Methods section to improve readability and organization.
Comments 3: Expalind the data extraction and software.
We have added a new subsection in the Methods section to describe the data extraction process and the software used for analysis.
Comments 4: Provide the comparison between results, and correlation analysis of data regarding the outcomes. Add a results of correlation
We appreciate your suggestion. In the initial version of the manuscript, a correlation analysis was included; however, after considering feedback from Reviewer 1, we have decided to remove it due to the limited number of time points (n = 4), which makes the analysis statistically unreliable and potentially misleading. Instead, we have strengthened the descriptive statistics in the Results section to provide a more comprehensive overview of the trends observed in antibiotic prescriptions, intervention rates, and acceptance outcomes. Additional tables and figures have been added to support this.
Comments 5: Discussion must longer with more data.
Thank you for your constructive suggestion. In the revised manuscript, we have expanded the Discussion section by incorporating a more detailed interpretation of the findings, including an in-depth discussion of the study’s limitations, the implications of ASP intervention trends, and the practical challenges of stewardship implementation in a real-world setting. We also included potential directions for future multicenter studies and ASP expansion with additional personnel. These revisions aim to provide a more comprehensive and data-supported discussion.
Comments 6: How do you explaine high acceptance rate ?What could be a reason for that?
We appreciate your insightful question. The high acceptance rate observed during the study period may be attributed to several factors. First, the ASP interventions were implemented through an internal electronic messaging system, which allowed prompt and clear communication between the ASP team and prescribing physicians. Second, the ASP recommendations were provided by board-certified infectious disease specialists, which may have increased the credibility and clinical trust in the suggestions. Lastly, many of the restricted antibiotics were used in severe or resistant infections, and the ASP team made efforts to approve prescriptions when clinically justifiable, which likely contributed to the high overall acceptance rate.
Comments 7: How do you explaine low rejection rate rate ?What could be a reason for that?
Thank you for your comment. The low rejection rate can be explained by the fact that most antibiotic prescriptions submitted for ASP approval were clinically appropriate, especially given that restricted antibiotics were often reserved for critically ill patients with resistant infections. Additionally, the ASP team aimed to support rather than hinder patient care, approving prescriptions that were justified by clinical context. Early communication and feedback with prescribers likely enhanced mutual understanding and reduced the need for rejection. Furthermore, borderline or ambiguous cases were often managed through direct consultation rather than formal rejection, contributing to the low observed rejection rate.
Comments 8: reference list must be lomger at least 30 references
Thank you for your suggestion. We have thoroughly revised the manuscript and expanded the reference list to include a total of 31 references, incorporating additional relevant literature on antimicrobial stewardship programs (ASPs), previous intervention studies, and recent national policy efforts in Korea. These references were added especially in the Introduction and Discussion sections to strengthen the context and support key findings of the study.
Comments 9: English must checked
Thank you for your comment. We have already had the manuscript professionally proofread by an English editing company specializing in academic manuscripts in Korea. However, we are willing to request additional professional editing if necessary to ensure the highest language quality.
Comments 10: Conclusion must be more focused on results
Thank you for the helpful suggestion. We revised the Conclusion section to more clearly highlight the study’s key findings, including the high acceptance rate, early downward trend in restricted antibiotic DOT, and feasibility of ASP implementation in a resource-limited secondary care setting. The revised conclusion emphasizes the observed outcomes and their potential implications for future practice and policy.
Comments 11: In article are missing information about ethical concerns and approval of study etc.
Thank you for your comment. We have added a statement regarding ethical approval and study oversight at the end of the Methods section.
Reviewer 3 Report
Comments and Suggestions for Authors
This manuscript carried out a single-center study analyzed ASP activity from January to April 2025. There are some questions I’d like the authors to answer:
- Please further explain the reason of data choosing for this single-center study, eg: Why did the authors choose data from January to April 2025? Is the time period sufficient for this study?
- Please indicate the logic of data analysis, eg: why did the authors display data and result monthly? Is it related to the treatment cycle?
- In figure 3, the X labels are different from previous figures.
- Please further indicate how to calculate the rejection rate, DOT and acceptance rate. It will be much improved to put all data in a summarized table and list the calculating formula.
Author Response
Comments 1: Please further explain the reason of data choosing for this single-center study, eg: Why did the authors choose data from January to April 2025? Is the time period sufficient for this study?
Thank you for this valuable comment. The data collection period (January to April 2025) corresponds to the initial four months following the launch of the ASP pilot program at our institution. This time frame was chosen to evaluate the early feasibility and impact of the newly implemented ASP in a real-world clinical setting. While we acknowledge that the limited observation period restricts long-term outcome analysis, we believe it is meaningful to share the initial results and implementation challenges during the formative phase of the program. This can serve as a foundation for future research with extended follow-up periods and broader generalizability.
Added Text (Discussion, Lines 149–152):
The four-month observation period (January to April 2025) reflects the early implementation phase of the ASP pilot project at this hospital. This initial timeframe was selected to assess the feasibility and preliminary outcomes of ASP interventions in real-world clinical practice.
Comments 2: Please indicate the logic of data analysis, eg: why did the authors display data and result monthly? Is it related to the treatment cycle?
Thank you for this important question. The data and results were presented on a monthly basis to reflect the operational nature of the ASP, which was implemented continuously and monitored in real time. Monthly analysis allowed us to observe trends over time, evaluate changes in prescription patterns, and assess the acceptability of stewardship interventions at regular intervals. While not directly related to a clinical treatment cycle, the monthly format was chosen to align with the hospital’s administrative and reporting cycles, which follow a monthly pattern for drug utilization and stewardship audits.
Comments 3: In figure 3, the X labels are different from previous figures.
Thank you for your careful observation. We have corrected the X-axis labels in Figure 3 to maintain consistency with the other figures. The revised version is included in the updated manuscript.
Comments 4: Please further indicate how to calculate the rejection rate, DOT and acceptance rate. It will be much improved to put all data in a summarized table and list the calculating formula.
Thank you for your comment. We have added the definitions and calculation methods for acceptance rate, rejection rate, and days of therapy (DOT) in the revised manuscript (Lines 93–95). To enhance clarity, the summary data have been organized in Supplementary Table S3. This tables provide monthly totals of restricted antibiotic prescriptions and DOTs for each agent.
Reviewer 4 Report
Comments and Suggestions for Authors
Please see attached file for comments. I commend you on your work for ASP--this is very important work. I feel you lack power and a sufficient sample size for inferential statistics, but the description of your program is very important. I would suggest just using descriptive statistics and describe the program as an early example of a program that should be replicated in many sites around the world.

Author Response
Please see attached file for comments. I commend you on your work for ASP--this is very important work. I feel you lack power and a sufficient sample size for inferential statistics, but the description of your program is very important. I would suggest just using descriptive statistics and describe the program as an early example of a program that should be replicated in many sites around the world.
Comments 1: Since the sample size is so small, I would suggest just providing descriptive statistics without additional inferential statistics. I would avoid using terms such as trend.
Thank you for your suggestion. In accordance with your comment, we have removed all inferential statistics, including correlation analyses, from the revised manuscript. Additionally, we have refrained from using the term “trend” and instead focused solely on descriptive statistics to present the findings more appropriately given the sample size.
Comments 2: This type of descriptive stats are good ways to demonstrate this data.
Thank you for the positive comment. We are pleased that the use of descriptive statistics in this section was found to be appropriate and effective for presenting the data.
Comments 3: Not statistically powered to have meaning at this point. Sample size and power are an issue.
Thank you for your comment. We agree that the sample size and study power limit the ability to draw statistically meaningful conclusions. Accordingly, we have revised the sentence to present a more cautious interpretation of the findings.
The sections marked with strikethrough have also been addressed and revised accordingly.
Round 2
Reviewer 3 Report
Comments and Suggestions for Authors
The authors have answered the questions legitimately.
Reviewer 4 Report
Comments and Suggestions for Authors
Minor edits in uploaded file
